# Association of First-Trimester Maternal Biomarkers with Preeclampsia and Related Maternal and Fetal Severe Adverse Events

**DOI:** 10.3390/ijms26146684

**Published:** 2025-07-11

**Authors:** Ana Camacho-Carrasco, Jorge Montenegro-Martínez, María Luisa Miranda-Guisado, Rocío Muñoz-Hernández, Rocío Salsoso, Daniel Fatela-Cantillo, Lutgardo García-Díaz, Pablo Stiefel García-Junco, Alfonso Mate, Carmen M. Vázquez, Verónica Alfaro-Lara, Antonio J. Vallejo-Vaz, Luis M. Beltrán-Romero

**Affiliations:** 1Internal Medicine, Infanta Elena Hospital, 21080 Huelva, Spain; 2Department of Clinical Biochemistry, Virgen del Rocío University Hospital, 41013 Sevilla, Spain; 3Internal Medicine, Virgen del Rocío University Hospital, 41013 Seville, Spain; 4Clinical Epidemiology and Vascular Risk, Instituto de Biomedicina de Sevilla, IBiS/Hospital Universitario Virgen del Rocío/CSIC/Universidad de Sevilla, 41013 Seville, Spain; 5Departamento de Medicina, Facultad de Medicina, Universidad de Sevilla, 41009 Sevilla, Spain; 6Investigación Clínica y Traslacional en Enfermedades Hepáticas y Digestivas–SeLiver, Instituto de Biomedicina de Sevilla, IBiS/Hospital Universitario Virgen del Rocío/CSIC/Universidad de Sevilla, 41013 Sevilla, Spain; 7Departamento de Fisiología, Facultad de Biología, Universidad de Sevilla, 41012 Sevilla, Spain; 8CIBEREHD, Centro de Investigación Biomédica en Red de Enfermedades Hepáticas y Digestivas, 28029 Madrid, Spain; 9Unidad de Medicina Maternofetal, Genética y Reproducción, Hospital Universitario Virgen del Rocío, 41013 Sevilla, Spain; 10Departamento de Cirugía, Facultad de Medicina, Universidad de Sevilla, 41009 Sevilla, Spain; 11Departamento de Fisiología, Facultad de Farmacia, Universidad de Sevilla, 41012 Sevilla, Spain; 12Centro de Investigación Biomédica en Red (CIBER) de Epidemiología y Salud Pública, Instituto de Salud Carlos III, 28029 Madrid, Spain

**Keywords:** preeclampsia, biomarkers, PlGF, sFlt-1, cell-free nucleic acids, microvesicles, screening for pregnancy endpoints, late-onset preeclampsia, early-onset preeclampsia

## Abstract

To assess the association between known (PlGF, sFlt-1, betaHCG, PAPPA) and novel (cell-free DNA, cfDNA, and total endothelial and platelet microvesicles, MVs) maternal blood biomarkers measured at the first trimester with the later development of preeclampsia (PE) and PE-related severe adverse events (SAE), we conducted a retrospective case–control study including women with an established diagnosis of preeclampsia (cases) and healthy pregnant women (controls). Biomarkers were measured from first-trimester blood samples stored in a hospital biobank. *A total of* 89 women, 54 women with PE and 35 controls were included. PlGF showed good performance for diagnosing overall preeclampsia (AUC: 0.71; 95% CI 0.59–0.82), early-onset preeclampsia (AUC 0.80; 95% CI 0.68–0.9) and fetal-neonatal SAEs (AUC: 0.73; 95% CI 0.63–0.84). Multivariate models including clinical variables, PlGF and other biomarkers showed good to very good discrimination for the development of PE, early-onset PE and fetal-neonatal SAEs (AUCs of 0.87, 0.89 and 0.79, respectively). Platelet-derived MVs were the best isolated biomarker for late-onset PE and, combined with systolic blood pressure, showed good discrimination (AUC: 0.81; 95% CI 0.71–0.92). For maternal SAEs, a model incorporating cfDNA and sFlt-1 provided excellent discrimination (AUC 0.92; 95% CI 0.82–1.00). Our findings suggest that multivariate models incorporating both clinical variables and first-trimester biomarkers may improve risk stratification for PE, especially for late-onset PE and for identifying women at risk of severe maternal or fetal complications. Notably, the inclusion of novel biomarkers such as cfDNA and MVs added value in clinical scenarios where the predictive performance of existing tools remains suboptimal.

## 1. Introduction

Preeclampsia (PE) is a relatively common hypertensive disorder of pregnancy [1] that affects 3% to 5% of all pregnancies [2,3] and represents a leading cause of maternal and perinatal morbidity and mortality [4,5]. PE may also result in long-term adverse health outcomes for both women and the offspring [1,6].

The clinical manifestations of PE usually become apparent in the second half of pregnancy; however, the pathogenic mechanisms leading to PE start developing during the first months of pregnancy. Therefore, early detection of pregnant women at risk of developing PE is of utmost importance, as it will allow a close monitoring of the pregnancy, the implementation of preventive strategies, and an early treatment when necessary [7]. Significant advances have been made by introducing first-trimester screening and preventive intervention with aspirin, but this approach neither foresees nor averts the majority of cases of PE [8].

PE can be classified according to the gestational age at diagnosis as early-onset PE (onset before 34 weeks of gestation) and late-onset PE (onset at or after 34 weeks of gestation) [9]. Early-onset PE is associated with significant morbidity and mortality, particularly of the fetus, but it is relatively uncommon (approximately 0.38% of pregnancies) compared to late-onset PE (approximately 2.72% of pregnancies) [10].

PE is a multifactorial and complex disorder characterized by abnormal placentation, endothelial dysfunction, metabolic dysfunctions and systemic inflammatory response. Recent research highlights the involvement of inflammatory and metabolic dysfunctions, contributing to the complex pathogenesis of the disease [11].

Several clinical risk factors for the development of PE have been identified, including a history of PE, chronic hypertension, diabetes, obesity, and advanced maternal age [8]. These clinical risk factors, ultrasound findings and maternal blood biomarkers have been proposed as early predictors (first trimester–preclinical stage) of PE [12]. Reasonably accurate prediction tools have been developed for early-onset PE, but predicting late-onset PE still remains an important challenge. On the other hand, clinical manifestations and the impact of PE (whether early or late onset) on maternal and fetal outcomes are variable, and women with either form of PE may develop severe complications [5,9,13]. Therefore, the improvement of the early prediction of PE, particularly in two scenarios, namely late-onset PE and PE that may develop severe adverse events (SAE), remains an unmet need.

Beyond placental secreted pro- and anti-angiogenic factors such as placental growth factor (PlGF) and soluble FMS-like tyrosine kinase-1 (sFlt-1), respectively, other novel biomarkers may play a role [8,14]. Among them, markers of endothelial dysfunction, vascular damage and cellular damage, such as cell microvesicles (MVs, including endothelial and platelet-cell-derived MVs) or cell-free DNA (cfDNA) [15,16,17,18].

In this study, we aimed to assess the association between already known maternal blood biomarkers of angiogenesis and novel biomarkers of endothelial dysfunction and cell damage measured in the first trimester of pregnancy with (i) a later diagnosis of PE, overall and stratified by early and late onset PE, and (ii) the development of maternal SAE and fetal outcomes. We also conducted multivariate models combining clinical variables and the studied biomarkers for the prediction of the development of PE and PE-related SAE.

## 2. Results

The study included 54 women with PE (23 with early-onset PE and 31 with late-onset PE) and 35 healthy control women who did not develop PE (Figure 1). The onset of PE occurred at 35.3 weeks [IQR 31.8–37.2] of gestation overall; the corresponding figures for PE women stratified by early- and late-onset PE were 31.6 [IQR 28.3–32.7] and 37.1 [IQR 35.7–38.3] weeks, respectively (*p* < 0.001). Gestational age at delivery was 36 weeks [IQR 34.0–38.0] in women with PE and 39 weeks [IQR 38.0–40.0] among control women (*p* = 0.001). Among patients with PE, gestational age at delivery was 34 [IQR 30–34] and 37 [IQR 37–38] weeks for those with early-onset and late-onset PE, respectively (*p* < 0.001).

Table 1 shows the characteristics of participants at the first trimester. Median age of women was similar in all groups (34.6 years [IQR 31.0–38.6]). Overall, women in the PE group were more frequently primiparous and tended to have a higher frequency of multiple gestation and pregnancy from in vitro fertilization (IVF), as well as slightly higher BP and platelet count. At the first trimester visit, 11 patients were on aspirin: 9 in the PE group and 2 in the control group. Over the course of pregnancy, one additional woman (PE group) started treatment with aspirin too.

Gestational age at which blood samples were collected was similar for both PE and control groups and for early- and late-onset PE groups: overall, 10.3 weeks [IQR 9.9–10.9] of gestation, Table 1. Levels of biomarkers in the first trimester are shown in Table 2.

### 2.1. Association of Biomarkers at First Trimester with a Later Diagnosis of PE

Women who later developed PE had lower levels of PlGF and β-hCG and higher levels of cfDNA in the first trimester compared to women with normal pregnancies (Table 2).

PlGF performed the best to predict the development of PE (AUC: 0.71, 95% IC 0.59–0.82) (Figure 2A). A cut-off of 36.3 pg/mL showed a sensitivity of 85% and specificity of 55%. Women with PlGF levels ≤ 36.3 pg/mL developed PE at an earlier gestational age than women with PlGF > 36.3 pg/mL (Figure 3A). Kaplan–Meier curves considering together participants with PE and controls, stratified by the PlGF cut-off of 36.3 pg/mL, showed that at week 30, 15% of women with a PlGF ≤ 36.3 and none of the women with PlGF > 36.3 had developed PE. At week 35, these figures were 40% and 15%, respectively.

cfDNA, β-hCG and the sFlt-1/PlGF ratio showed an AUC of 0.65, 0.70 and 0.65, respectively (all *p* < 0.05). Neither MVs nor sFlt-1 in the first trimester were significantly associated with a later diagnosis of PE.

A multivariate model including primiparous status, PlGF, cfDNA and β-hCG showed an AUC of 0.87 (95% CI 0.80–0.95) for the prediction of PE (Figure 4A).

### 2.2. Association of Biomarkers at First Trimester with Early-Onset PE

Women who developed early-onset PE had lower levels of PlGF, PAPP-A and total and platelet MVs in the first trimester compared to women with late-onset PE (Table 2). Kaplan–Meier curves for the PE group specifically, stratified by the PlGF cut-off of 36.3 pg/mL, are shown in Figure 3. Women with PlGF levels ≤ 36.3 pg/mL tended to develop early-onset PE more frequently, whereas all women with PlGF levels > 36.3 pg/mL developed PE at week 34 or later (i.e., late-onset PE). At week 30, almost 20% of women with a PlGF ≤ 36.3 and none of the women with PlGF > 36.3 had developed PE. At week 35, these figures were almost 60% and 15%, respectively (Figure 3B).

Compared to women who did not develop PE, women with early-onset PE showed lower levels of PlGF, β-hCG and PAPP-A in the first trimester (Table 2).

For the performance of biomarkers to predict the development of early-onset PE, we compared, in the first trimester, women who developed early-onset PE vs. the rest of the women (controls and women who developed late-onset PE; in a clinical setting, it will be unknown who will develop PE; thus, in predicting early-onset PE vs. everyone else, everyone else should include healthy controls and those developing late-onset PE). PlGF performed the best for the diagnosis of early-onset PE with an AUC of 0.76 (95% CI 0.66–0.87), Figure 2B. sFlt-1/PlGF ratio and PAPP-A presented limited value for discrimination with AUCs of 0.69 and 0.68, respectively. sFlt-1 was not associated with early-onset PE. A multivariate model including primiparous status, in vitro fertilization (IVF), PlGF and PAPPA showed a very good discrimination for early-onset PE (AUC 0.89; 95% CI 0.82–0.96) (Figure 4B). When the biomarkers PlGF and PAPP-A were removed and only the clinical variables were retained, the AUC decreased to 0.77 (95% CI: 0.65–0.89).

### 2.3. Association of Biomarkers at First Trimester with Late-Onset PE

Compared to women who did not develop PE, women with late-onset PE showed higher levels of cfDNA and platelet MVs and lower levels of β-hCG (and borderline statistically significant lower PlGF, *p* = 0.052) measured at the first trimester (Table 2).

However, when comparing, at first the trimester, women with late-onset PE vs. the rest of the women (controls and women who developed early-onset PE; same rationale as above), none of the biomarkers showed a good discrimination value in isolation for a late-onset PE diagnosis. The AUC for a diagnosis of late-onset PE for platelet MVs was 0.69 (Figure 2C). However, when combined with clinical variables, a multivariate model including platelet MVs and systolic blood pressure showed an AUC of 0.81 (95% CI 0.71–0.92) for late-onset PE (Figure 4C). We generated an ROC curve using the clinical variable alone—systolic blood pressure—which showed limited discriminative performance (AUC 0.61, 95% CI: 0.49–0.74, *p* = 0.059).

### 2.4. Maternal and Fetal-Neonatal Severe Adverse Events

There were 50 SAE: 43 fetal-neonatal and 7 maternal. Details of events are shown in Table 3. There was a significant association between early-onset PE and fetal-neonatal SAE (*p* < 0.001). No association was found for early or late-onset PE and the development of maternal SAE (*p* = 0.13 and *p* = 0.19, respectively) (though we need to consider that the number of maternal SAE was only 7).

Table 4 shows the biomarkers measured in the first trimester stratified by those women who later during pregnancy had or did not have SAE. Women who experienced SAE had significantly higher levels of sFlt-1, cfDNA, β-hCG and PAPP-A in the first trimester than women who did not experience SAE.

PlGF showed an AUC of 0.74 (95% CI 0.64–0.85) for the occurrence of the combined maternal–fetal outcome (Figure 2D). A multivariate model including IVF, PlGF and maternal age showed an AUC of 0.78 (95% IC 0.68–0.88) (Figure 4D). After removing PlGF, maternal age lost statistical significance, supporting the discriminative value of PlGF in this model.

PlGF discriminated well for the fetal-neonatal SAE, with an AUC of 0.73 (95% CI 0.63–0.84) (Figure 2E). A model including multiple gestation, PlGF and PAPP-A showed an AUC of 0.79 (95% CI 0.69–0.89) (Figure 4E).

In case of maternal SAE, sFlt-1 showed a good discrimination with an AUC of 0.80 (95% CI 0.56–1.00) (Figure 2F). The AUC for β-hCG, PAPP-A and cfDNA were 0.73, 0.73 and 0.73, respectively. A multivariate model including sFlt-1 and cfDNA showed an excellent discrimination value for maternal SAE (AUC 0.92; 95% CI 0.82–1.00) (Figure 4F).

In addition to ROC curve analysis, we explored the strength of association between key biomarkers and adverse outcomes using odds ratios (ORs) with 95% confidence intervals. PLGF levels > 36 pg/mL were associated with significantly lower odds of adverse fetal events compared to levels ≤ 36 pg/mL (OR: 0.25, 95% CI: 0.09–0.69; *p* = 0.007). A similar association was observed when considering combined maternal and fetal adverse events (OR: 0.27, 95% CI: 0.10–0.70; *p* = 0.008). On the other hand, elevated sFLT1 concentrations (>1858 pg/mL) were associated with an increased OR of maternal adverse events (OR: 29.14, 95% CI: 3.25–261.38; *p* = 0.003). There was also a non-significant trend toward higher maternal odds with cfDNA levels > 1162 ng/mL (OR: 8.06, 95% CI: 0.93–69.99; *p* = 0.059). No statistically significant associations were found between cfDNA or sFLT1 levels and fetal or combined outcomes, nor between PLGF and maternal adverse events. The full results for the association of ORs with SAEs are provided in Appendix A.

There was no relationship between aspirin treatment in the first trimester or during the course of pregnancy with PE or with maternal–fetal adverse outcomes.

## 3. Discussion

Our study shows that biomarkers measured at the first trimester were associated with the later occurrence of PE-related SAEs. We also observed a distinct biomarker profile associated with fetal events (fetal syndrome) or maternal events (maternal syndrome). Interestingly, placental-related biomarkers such as PlGF were associated with fetal events, whereas endothelial dysfunction and cell stress biomarkers (cfDNA and sFlt-1) were linked to maternal events. Multivariate models incorporating these biomarkers and clinical features showed an acceptable discrimination for the occurrence of fetal events (a model including PlGF, PAPP-A and multiple gestation status showed an AUC of 0.79) and an excellent discrimination for maternal adverse events (a model including sFlt-1 and cfDNA showed an AUC of 0.916).

PlGF was the best isolated predictor to screen at the first trimester for the later diagnosis of PE, particularly for early-onset PE. A cut-off value of 36.3 pg/mL showed 85% sensitivity at 55% specificity for PE diagnosis. Around 40% of pregnant women with PlGF levels < 36 pg/mL at the first trimester developed early-onset PE compared with less than 10% of those with PlGF levels > 36 pg/mL. A multivariate model using primiparous status, IVF, PlGF and PAPP-A showed a very good discrimination for early-onset PE diagnosis with an AUC of 0.89. Despite our work not identifying any strong isolated predictor for late-onset PE, we found that a multivariate model including platelet MVs and systolic blood pressure had a good discrimination value (AUC 0.81).

There is limited evidence regarding the prediction of PE-related maternal or fetal SAE, and almost none using early predictors in the first trimester. Some studies have proposed some clinical and laboratory variables at diagnosis and follow-up of women with PE, such as blood pressure, hemoglobin, lactate dehydrogenase, platelet counts or plasma creatinine, for the prediction of maternal outcomes [13,19,20]. Other studies identified the sFlt-1-to-PlGF ratio as a short-term predictor of maternal and fetal outcomes in women with suspected PE [21,22]. Sun et al. found that higher levels of sFlt-1 among patients with severe PE were correlated with adverse pregnancy outcomes (maternal and fetal) [23]. However, several of these features are indicators of already ongoing organ damage, such as hemoglobin, lactate dehydrogenase or creatinine. On the other hand, all of these studies have been conducted in the last half of pregnancy in women with suspected or diagnosed PE, and therefore, they inform short-term management decisions but leave little room for longer preventive measures. Our study suggests that certain maternal blood biomarkers measured during the first trimester of pregnancy may help identify subgroups of women at increased risk for PE-related SAEs, beyond their association with the diagnosis of PE itself. This may be particularly crucial, as previous categorizations of PE as severe and non-severe inadequately stratify individuals who might develop severe complications, and, consequently, this stratification approach is no longer endorsed by clinical guidelines [5].

We considered the possibility that the observed associations between first-trimester biomarkers (e.g., sFlt-1 and cfDNA) and maternal severe adverse events (SAEs) might reflect their link with preeclampsia (PE). However, adjusting for PE in this context would not be methodologically appropriate, as PE develops after the exposure (biomarker levels) and may lie on the causal pathway to adverse outcomes. In this scenario, PE acts as a mediator, not a confounder, and adjustment could introduce overadjustment bias. Moreover, one of the objectives of our study was to evaluate whether first-trimester markers could help identify, among patients at risk for or developing PE, those more likely to experience severe complications. This supports our choice to retain unadjusted associations for these specific analyses.

Regarding early-onset PE, our results are consistent with previously published data. However, they are particularly promising for late-onset PE, where previous predictive models had been suboptimal. A prior systematic review identified PlGF as the best early predictor of PE with 65% sensitivity at 89% specificity [24]. PE risk scores based on clinical factors have been proposed by clinical practice guidelines [25,26], but these models have shown modest performance with sensitivity around 40% for the prediction of early-onset PE and even poorer for all PE cases, including late-onset ones [27]. A first-trimester screening algorithm including maternal features, mean arterial blood pressure, mean uterine artery resistance and PlGF levels has been developed and validated for accurately predicting early-onset PE [28]. However, the accuracy in predicting late-onset disease is significantly lower, as it identifies only 44% of women who will develop PE at ≥37 weeks of gestation [27]. Other protocols using maternal features and Doppler ultrasonography parameters have shown good performance for late-onset PE when applied in the third trimester but had poor performance when used early in pregnancy, when preventive measures may have a larger impact in preventing the later development of PE. We found promising data regarding early prediction of late-onset PE, a form of the disease for which first-trimester prediction remains suboptimal. A model including platelet MVs and systolic blood pressure achieved a good discrimination for late-onset PE (AUC 0.81).

Our study contributes novel evidence supporting the potential clinical relevance of emerging maternal blood biomarkers such as cell-free DNA and platelet-derived MVs. These markers, particularly when incorporated into multivariate models with traditional clinical variables, demonstrated added value particularly in two challenging but clinically important scenarios for which previously described biomarkers have limited predictive performance: the prediction of late-onset preeclampsia and the early identification of patients at risk for maternal SAEs. Late-onset PE, despite being the most common type of PE, remains difficult to predict using current screening tools. In our study, platelet-derived MVs improved the performance of models for late-onset PE when added to clinical variables. Similarly, cfDNA—alone and in combination with sFlt-1—showed strong discriminative ability for predicting maternal SAEs, for which early predictive markers are lacking. These findings highlight the potential of novel biomarkers to fill important gaps in current risk stratification strategies and support their inclusion in future prospective validation studies.

Importantly, these biomarkers and models would be applied in the first trimester of pregnancy, allowing for early preventive measures and close monitoring, i.e., they could have a more significant and larger impact on prognosis and potentially lead to avoiding adverse events. While these findings are promising, we acknowledge that the models presented in this study are exploratory and require external validation in prospective cohorts.

On the other hand, our findings support the notion that distinct pathophysiological mechanisms may underlie the development of early- versus late-onset PE and in the so-called PE-related fetal syndrome vs. maternal syndrome. This heterogeneity justifies the development of separate predictive models tailored to each outcome. While fetal events are associated with biomarkers of placental dysfunction (PlGF and PAPP-A), likely reflecting an inadequate vascularization and consequently abnormal fetal perfusion, maternal events may be related to endothelial dysfunction and cell stress (sFlt-1 and cfDNA). sFlt-1 is a well-known biomarker for the diagnosis of suspected PE, and it has been associated with antiangiogenic effects and abnormal placentation. However, it can also cause endothelial dysfunction and oxidative stress [29,30]. Outside of pregnancy, sFlt-1 has been linked to heart failure and atherosclerotic cardiovascular disease, supporting the notion that sFlt1-mediated vascular effects could be a nexus between this biomarker and adverse maternal events [31,32]. These findings could assist in designing novel and more personalized diagnostic and preventive strategies aiming at targeting the specific affected pathways in each case (early vs. late-onset PE and fetal vs. maternal syndrome).

### Strengths and Limitations

One of the main strengths of our study is that biomarker measurements were performed on first-trimester blood samples collected during routine clinical care, supporting the feasibility of integrating these assessments into real-world clinical workflows. Additionally, the study focused on clinically relevant but often underexplored scenarios, such as the early identification of late-onset PE and the prediction of severe maternal and fetal adverse outcomes. The inclusion of both established biomarkers (e.g., PlGF, sFlt-1) and novel markers of endothelial dysfunction and cell damage (e.g., cfDNA, microvesicles) allowed us to explore their complementary value. Moreover, the development of multivariable models combining biomarkers with easily accessible clinical parameters enhances the translational applicability of our findings.

One of the main side limitations of our study is the loss of 43 eligible pregnancies due to the unavailability of stored biological samples. These losses were primarily due to pre-analytical and logistical issues unrelated to the outcome, such as insufficient volume, degradation, or mislabeling during storage. Although non-differential in nature, this loss could reduce statistical power and introduce selection bias. Despite these losses, our final sample size remained close to the a priori estimated target based on formal power calculations. Another potential limitation is the non-uniform use of low-dose acetylsalicylic acid (ASA), which was prescribed selectively according to clinical criteria based on existing obstetric guidelines. While this introduces a degree of therapeutic heterogeneity, ASA use was not significantly associated with the occurrence of preeclampsia or adverse outcomes in our sample. Therefore, its influence on the associations observed in this study is likely to be limited. We lack data regarding first-trimester Doppler ultrasonography, which could have affected the components and performance of the proposed multivariate models, although presumably enhancing them. Also, it could have been informative to measure biomarker levels later in pregnancy as well and to evaluate changes over the pregnancy course and their predictive relevance. Finally, the analysis of microvesicles in our study was conducted using biobanked samples obtained under routine clinical conditions, which did not allow for strict control of processing times. Although this may have affected the accuracy of absolute MV quantification, all samples—cases and controls—were collected, stored, and processed under the same conditions. This ensures internal comparability and supports the exploratory evaluation of their relative discriminatory performance. Nonetheless, these findings should be interpreted cautiously and confirmed in prospective studies using standardized protocols for MV handling and analysis.

## 4. Material and Methods

### 4.1. Study Design and Population

We conducted a retrospective study including women with an established diagnosis of PE (cases) and healthy pregnant women as controls. All participants were recruited after delivery to ensure the presence or absence of a definite diagnosis of PE. The inclusion criteria for the group of women with PE (hereinafter referred to as the “PE group”) consisted of women aged 18–50 years old with a confirmed diagnosis of PE, eclampsia or HELLP syndrome. The control group consisted of women aged 18–50 years who did not develop PE, eclampsia, HELLP syndrome, or other forms of hypertensive disorders of pregnancy (gestational hypertension or chronic hypertension) during the pregnancy or in the postpartum period. The exclusion criteria for both groups included the absence of blood samples from the first trimester stored in the hospital biobank to measure the study biomarkers; the presence of any inflammatory or autoimmune disease or use of immunomodulatory drugs; previous evidence of atherosclerotic vascular disease (coronary artery disease, cerebrovascular disease, or peripheral artery disease) or chronic diseases that may affect cell damage or endothelial function biomarkers (e.g., diabetes mellitus at the first trimester visit, chronic kidney disease, or HIV); and smoking.

All eligible participants were identified and invited to participate during their postpartum hospitalization, after completion of the pregnancy. At that time, written informed consent was obtained for the use of their clinical and perinatal data and for accessing previously collected biological samples. These samples had been drawn and stored during the first-trimester routine antenatal visit, which is universally offered to all pregnant women in our healthcare system. This design enabled retrospective analysis of early biomarkers in pregnancies with well-documented maternal and neonatal outcomes. Women with PE were recruited consecutively from July 2019 to July 2021; control pregnant women were recruited consecutively over several days in the obstetric ward. The study was conducted at the Virgen del Rocío University Hospital, a tertiary hospital in Seville, Spain.

Out of 137 candidates (65 control women and 72 women with PE), 29 control women and 14 patients with PE had no available sera from the first-trimester screening visit in the hospital serum bank, 1 woman had an estimated gestational age of 17 weeks +3 days at the time of blood sample collection, 1 patient had no clinical information in our electronic clinical records system, and 3 cases presented other exclusion criteria. Therefore, 89 women (35 healthy controls and 54 women with PE) were finally included in the study (Figure 1).

### 4.2. Sample Size

The sample size calculation was based on previous data from a pilot study [15]. Assuming a clinically relevant difference of 10% in mean total circulating DNA levels between groups, a power of 80% to detect differences under the null hypothesis (H_0_: μ_1_ = μ_2_), and a two-tailed Student’s *t*-test for independent samples with an alpha of 0.05, a minimum of 142 subjects were required (82 with preeclampsia and 60 controls). This calculation was based on an expected proportion of 27% in the experimental group, a reference group mean of 335 GE/mL, a standard deviation of 50 units, and an anticipated sample loss of up to 50%. The final sample size was close to this target but was partially affected by missing samples due to technical limitations.

### 4.3. Events Definitions

A diagnosis of PE, eclampsia or HELLP syndrome was established according to the definitions of the American College of Obstetricians and Gynecologists [25]. PE was classified as early or late onset PE if the clinical onset occurred before 34 weeks of gestation or at or after 34 weeks of gestation, respectively [9].

Maternal SAE outcome was defined as the composite of maternal death; maternal admission to intensive care unit (ICU); development of eclampsia; disseminated intravascular coagulation; acute renal failure; acute liver failure; and severe neurological complications other than eclampsia (stroke or posterior reversible encephalopathy syndrome). Admission to a monitored unit different from the ICU (i.e., a recovery unit) and non-severe neurological complications such as headache or isolated visual disturbances were not included. Fetal and neonatal SAE outcomes included fetal death; intrauterine growth retardation; neonatal admission to ICU; and neonatal death before 30 days of age. The composite fetal or maternal SAE outcome was considered positive if a maternal or fetal/neonatal SAE occurred.

### 4.4. Data Sources and Quality Control

Clinical, obstetric, and neonatal data were obtained from the hospital’s electronic medical records, which are used routinely in patient care. These records include prospectively collected variables such as blood pressure measurements, delivery details, and maternal–fetal outcomes, all of which are entered by attending clinicians as part of standard care.

In contrast, data on maternal biomarkers were generated specifically for this study. Biomarker measurements were performed using stored first-trimester serum or plasma samples retrieved from the institutional biobank. The resulting values were then entered into a secure research database for statistical analysis. Only cases with complete data for the relevant variables were included in each analysis.

### 4.5. Sample Collection, Storage and Biomarker Analyses

Samples from the participating women were obtained from the serum bank of the Virgen del Rocío University Hospital. The sera had been obtained from the routine first-trimester blood sample collection for blood testing as per routine clinical practice. Maternal blood had been collected through venipuncture in tubes without anticoagulant. Subsequently, serum samples were centrifuged at 2000× *g* after clot retraction, and the supernatant was stored at −80 °C.

The methods for the measurement of biomarkers are described in detail in the Appendix A. Briefly, sFlt-1 and PlGF levels were determined using electrochemiluminescence immunoassays (ECLIA). Total cell-free DNA (cfDNA) (fetal plus maternal) was measured from the sera using the QIAamp Blood Mini Kit (Qiagen, UK). β-hCG and PAPP-A concentrations were determined using an electrochemiluminescence technique on a Cobas 6000 analyzer (Roche Diagnostics Penzberg, Germany). Cell microvesicles (MVs) were determined using flow cytometry (BD LSR Fortessa; BD Biosciences, San Jose, CA, USA). Total microvesicles were identified by annexin V positivity (AV^+^); endothelial MV were identified with the markers AV^+^, CD31^+^ and CD4^−^; and platelet MVs were identified with markers AV^+^, CD31^+^ and CD41^+^.

### 4.6. Statistical Analysis

Firstly, we performed a descriptive analysis of PE cases and controls and of PE patients stratified by early and late-onset PE, and we conducted between-group comparisons. Details are described in the Appendix A.

The degree of association between the variables of interest and the outcomes was estimated through univariate and multivariate logistic regression with estimation of odds ratios (OR) and corresponding 95% confidence intervals (95% CI). Receiver operating characteristic (ROC) curves were generated to assess the discriminatory capacity of biomarkers and of multivariable models for the outcomes of interest. We considered an area under the ROC curve (AUC) of 0.9 or higher as excellent, 0.8–0.9 as good, 0.7–0.8 as acceptable, 0.6–0.7 as fair and <0.6 as inadequate [14]. Different cut-off points were determined to assess sensitivity, specificity, positive and negative predictive values, accuracy, and their 95% CI.

For multivariate models, all studied biomarkers and relevant clinical variables (based on clinical risk factors mentioned in guidelines such as NICE [26]), all assessed at the first trimester visit, were included as follows: IVF, elevated systolic blood pressure (>140 mmHg), elevated diastolic blood pressure (>90 mmHg), multiple pregnancy, obesity (BMI > 30 kg/m^2^), maternal age (above vs. below 40 years), prior gestational hypertension or PE in previous pregnancies, primiparous status, and aspirin therapy in the first trimester. Given the exploratory nature of the study and the distinct clinical outcomes assessed (early- and late-onset PE, maternal and fetal SAEs), we developed separate multivariate models for each outcome. This approach was based on the hypothesis that different biological pathways may underlie each condition, and therefore distinct combinations of clinical and biochemical predictors may be required to optimize discrimination.

To assess the incremental value of biochemical biomarkers, we evaluated the discriminative performance of multivariable models built with clinical variables alone and then compared them to the full models that also included biochemical markers. ROC curves and AUC values were calculated for each configuration. In cases where only one binary clinical variable was available, no separate ROC curve was generated due to the limited interpretability of dichotomous predictors in this context.

Time of gestation free of PE (until onset of PE) was depicted graphically using Kaplan–Meier curves stratified by selected biomarkers cut-offs as detailed in the results section.

Tests were two-sided, and statistical significance was set at *p* < 0.05. Data were analyzed using IBM SPSS Statistics 29.0.1.0 (171).

### 4.7. Ethics

This study was approved by the Research Ethics Committee of the Virgen del Rocío University Hospital, Seville, Spain (PI-0456-2018). All participants provided written informed consent for their participation in the study.

## 5. Conclusions

Our study indicates that multivariable models integrating both established and novel first-trimester biomarkers demonstrate acceptable to very good discriminative performance in relation to the subsequent development of preeclampsia and its associated severe maternal and fetal adverse outcomes. Notably, our findings provide valuable new evidence in two areas where early-pregnancy prediction remains an unmet clinical need: the identification of late-onset preeclampsia, which has traditionally shown poor performance in screening models, and the prediction of severe maternal and fetal-neonatal adverse outcomes, for which data remain scarce.

Interestingly, our findings reveal that placental biomarkers, such as PlGF and PAPP-A, are more closely associated with early-onset PE and fetal complications, whereas markers of endothelial dysfunction and cell stress are more closely linked to maternal adverse outcomes (cfDNA and sFlt-1) and late-onset PE (platelet microvesicles). These results support the hypothesis of distinct underlying pathophysiological pathways between PE subtypes and their associated syndromes.

All models were developed using biomarkers and clinical information obtained during routine first-trimester care, reinforcing the feasibility of implementing such tools in real-world pregnancy monitoring programs. If externally validated, these models could contribute to precision medicine and personalized strategies for early diagnosis of PE and related severe adverse events, monitoring, and timely preventive interventions—particularly in pregnancies at elevated risk of severe preeclampsia-related adverse outcomes.

These results represent a promising step toward improving maternal and perinatal prognosis through stratified, early-risk assessment, and they provide a foundation for future prospective validation of the results in larger and diverse cohorts.

## Figures and Tables

**Figure 1 ijms-26-06684-f001:**
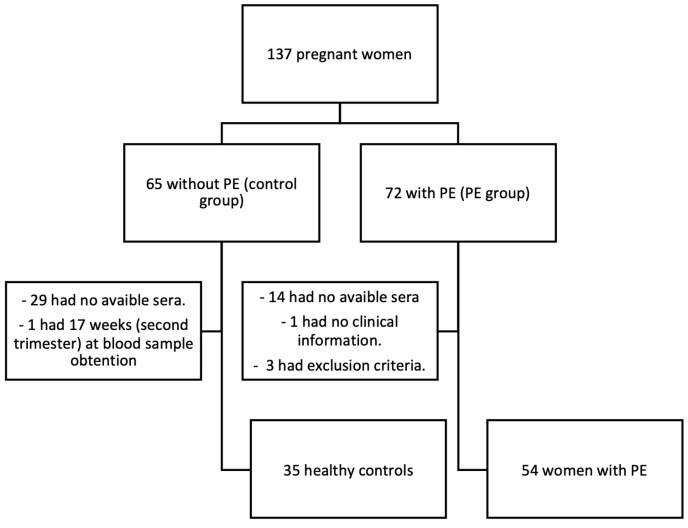
Inclusion of participants flowchart.

**Figure 2 ijms-26-06684-f002:**
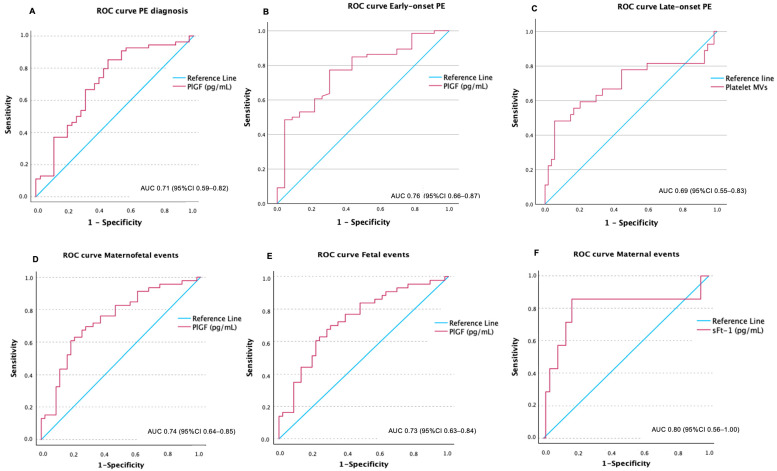
AUC of best performers for prediction of preeclampsia (**A**), early-onset preeclampsia (**B**), late-onset preeclampsia (**C**), maternal–fetal events (**D**), fetal events (**E**) and maternal events (**F**).

**Figure 3 ijms-26-06684-f003:**
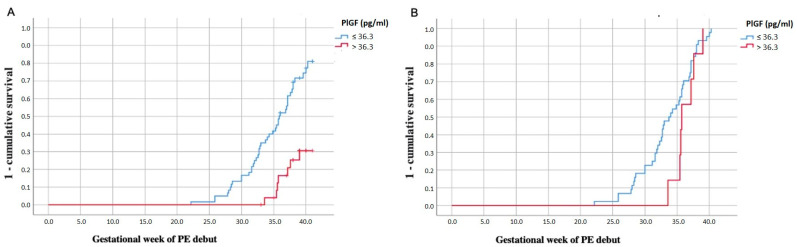
Kaplan–Meier curves for preeclampsia for the control and preeclampsia groups together (**A**) and for the preeclampsia group alone stratified by early- and late-onset preeclampsia (**B**).

**Figure 4 ijms-26-06684-f004:**
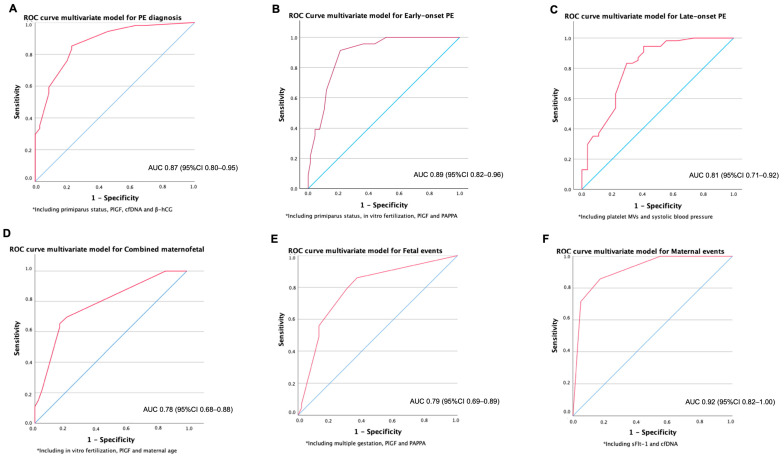
AUC of models for prediction of preeclampsia (**A**), early-onset preeclampsia (**B**), late-onset preeclampsia (**C**), maternal–fetal events combined (**D**), fetal-neonatal events (**E**) and maternal events (**F**).* Variables included in each multivariate model.

**Table 1 ijms-26-06684-t001:** Characteristics of participants at first trimester.

	Women Stratified Based on Whether They Developed or Did Not Develop Preeclampsia			Women Who Later Developed Preeclampsia, Stratified by Time of Diagnosis		
	Women Who Did Not Develop Preeclampsia	Women Who Developed Preeclampsia	*p*-Value	Early-Onset Preeclampsia	Late-Onset Preeclampsia	*p*-Value
	n = 35	n = 54	--	n = 23	n = 31	--
Age, years	34.45 (32.0–39.4)	34.6 (30.0–37.8)	0.383	34.87 (31.7–36.9)	34.54 (29.7–38.2)	0.923
Gestational week at blood sample obtention, weeks	10.57 (10.2–11.2)	10.22 (9.9–10.7)	**0.07**	10.2 (9.9–10.7)	10.3 (9.9–10.9)	1.0
Multiple gestation	1 (2.9%)	6 (11.1%)	0.238	3 (13.0%)	3 (9.7%)	0.697
Pregnancy from IVF	2 (5.7%)	11 (20.4%)	0.056	8 (34.8%)	3 (9.7%)	**0.024**
Primiparous mothers	5 (14.3%)	26 (48.1%)	**0.001**	15 (65.2%)	11 (35.5%)	**0.031**
Number of prior gestations	2.0 (2.0–3.0)	2.0 (1.0–2.0)	**0.022**	1.0 (0.0–1.0)	1.0 (0.0–2.0)	**0.031**
Prior miscarriage/abortion	11 (31.4%)	20 (37.0%)	0.587	4 (17.4%)	16 (51.6%)	**0.010**
Number of prior miscarriages/abortions	0.0 (0.0–1.0)	0.0 (0.0–1.0)	0.465	0.0 (0.0–0.0)	1.0 (0.0–1.0)	**0.026**
Preeclampsia in prior gestation(s)	1 (2.9%)	2 (3.7%)	1.000	0 (0.0%)	2(6.5%)	0.214
Fetal sex, male	16 (45.7%)	28 (51.9%)	0.572	12 (52.2%)	16 (51.6%)	0.967
If 2 fetuses, fetal sex of the 2nd fetus, male	1 (16.7%)	5 (83.3%)	1.000	2 (40%)	3 (60%)	0.273
If 3 fetuses, fetal sex of the 3rd fetus, male	1 (100%)	--	--	--	--	--
Body mass index, kg/m^2^	24.3 (21.4–27.8)	24.8 (22.3–31.4)	0.090	25.2 (22.7–30.8)	24.6 (22.3–35.8)	0.871
Systolic blood pressure, mmHg	110.0 (100.0–117.0)	111.5 (108.3–125.0)	**0.006**	115.0 (109.0–120.0)	111.0 (106.0–130.0)	0.705
Diastolic blood pressure, mmHg	62.0 (60.0–70.0)	70.0 (60.0–79.3)	**0.004**	70.0 [60.0–75.0]	70.0 [60.0–80.0]	0.691
Treatment with aspirin	2 (5.7%)	10 (18.5%)	0.084	5 (21.7%)	5 (16.1%)	0.6
Treatment with heparin	2 (5.7%)	2(3.7%)	0.644	1 (4.3%)	1 (3.2%)	0.829
Hb, g/dL	12.81 ± 1.0	12.88 ± 1.0	0.453	12.94 ± 1.1	12.85 ± 1.0	0.774
Platelets, 10^3^/mL	255 (201–295)	279 (247–302)	0.097	261 (219–302)	284 (255–304)	0.351

Legend for Table 1. Data are expressed as absolute and relative frequencies (n, %), mean ± SD, or median (25th–75th percentile). Hb, hemoglobin; IVF, in vitro fertilization.

**Table 2 ijms-26-06684-t002:** Biomarkers measured at the first trimester stratified by women who later developed or did not develop pre-eclampsia and by those with early-onset or late-onset PE.

	Women Who Did Not Develop Preeclampsia	Women Who Developed Preeclampsia	*p*-Value for Control vs. PE	Early-Onset PE	Late-Onset PE	*p*-Value for Early vs. Late-Onset PE	*p*-Value for Control vs. Early-Onset PE	*p*-Value for Control vs. Late-Onset PE
	n = 35	n = 54		n = 23	n = 31	--		
sFlt-1, pg/mL	1366 [1221–1857]	1261 [1003.6–1776.5]	0.320	1090 [909–1802]	1366 [1144–1768]	0.349	0.190	0.644
PlGF, pg/mL	37.12 [23.97–50.4]	25.94 [17.3–34.6]	**<0.001**	19.26 [14.86–28.92]	29.13 [21.48–36.21]	**0.005**	**<0.001**	0.052
sFlt-1/PlGF	42.32 [32.7–50.8]	50.09 [38.8–84.4]	**0.015**	61.67 [40.79–110.5]	48.39 [36.8–58.94]	**0.049**	**0.005**	0.156
β -hCG, ng/mL	61 [38.5–77.7]	36 [25–63.8]	**0.02**	35.9 [27.6–64.8]	37 [23.9–63.4]	0.937	0.007	**0.010**
PAPP-A, mUI/mL	1.77 [0.78–2.38]	1.01 [0.61–2.52]	0.248	0.7 [0.45–1.53]	1.65 [0.73–3.35]	**0.037**	**0.013**	0.807
cfDNA, ng/mL	849 [512–1580]	1210 [761–2060]	**0.020**	1170 [715–1910]	1580 [777–2240]	0.588	**0.117**	**0.022**
Total MVs *, (U/µL)	4775 [2781–7216]	5023 [2475–13,841]	0.540	3268 [1808–6067]	7445 [3103–18,584]	**0.030**	0.259	0.065
Endothelial MVs *, (AU/µL)	28.16 [17.9–44.12]	45.29 [8.2–327.1]	0.373	26.9 [8.12–245.6]	93.2 [12.5–339]	0.254	0.667	0.089
Platelet MVs *, (U/µL)	669.3 [411.6–1282.8]	829.1 [331.7–4686.8]	0.259	495.9 [300–833.9]	1512.8 [668.5–7697]	**0.023**	0.291	**0.010**

Legend for Table 2. Data are expressed as mean ± SD or median (25th–75th percentile). AV, annexin V; β-hCG, beta subunit of human chorionic gonadotropin; cfDNA, cell-free DNA; MVs, microvesicles; PAPP-A, pregnancy-associated plasma protein A; PlGF, placental growth factor; sFlt-1, soluble FMS-like tyrosine kinase-1. * Serum available in 81 women. Total microvesicles AV^+^; endothelial microvesicles AV^+^ CD31^+^ CD41^−^; platelet microvesicles AV^+^ CD31^+^ CD41^+^.

**Table 3 ijms-26-06684-t003:** Severe maternal, fetal and neonatal adverse events in the control group and in women with PE (overall, and by early- and late-onset PE).

	Women Stratified Based on Whether They Developed or Did Not Develop Preeclampsia			Women Who Later Developed Preeclampsia, Stratified by Moment of Diagnosis		
	Women Who Did Not Develop Preeclampsia	Women Who Developed Preeclampsia	*p*-Value	Early-Onset PE	Late-Onset PE	*p*-Value
	n = 35	n = 54	--	n = 23	n = 31	--
Composite maternal–fetal outcome	4 (11.4%)	42 (77.8%)	**<0.001**	21 (91.3%)	21 (67.7%)	**0.039**
Maternal outcomes	--	--	--	--	--	--
Composite maternal outcome	0	7 (13%)	**0.039**	2 (8.7%)	5 (16.1%)	0.685
Individual events:	--	--	--	--	--	--
Non-severe neurological symptoms	0	12 (22.2%)	**0.001**	7 (30.4%)	5 (16.1%)	0.211
Severe neurological complications	0	0	--	0	0	--
Eclampsia	0	0	--	0	0	--
Acute kidney failure	0	5 (9.3%)	0.152	2 (8.7%)	3 (9.7%)	1.000
Acute liver failure	0	3 (5.6%)	0.276	1 (4.3%)	2 (6.5%)	1.000
Disseminated intravascular coagulation	0	1 (1.9%)	1.000	0	1(3.2%)	1.000
Admission to monitored Unit	11 (31.4%)	44 (81.5%)	**<0.001**	21 (91.3%)	23 (74.2%)	0.161
Admission to intensive care unit	0	1 (1.9%)	1.000	0	1(3.2%)	1.000
Maternal death	0	0	--	0	0	--
Other maternal events:	--	--	--	--	--	--
Placental abruption	1(2.9%)	3 (5.6%)	1.000	1 (4.3%)	2 (6.5%)	1.000
Fetal outcomes	--	--	--	--	--	--
Composite fetal outcome	4(11.4%)	39 (72.2%)	**<0.001**	21 (91.3%)	18 (58.1%)	**0.007**
Individual events:	--	--	--	--	--	--
Intrauterine growth restriction	0	24 (44.4%)	**<0.001**	14 (60.9%)	10 (32.3%)	**0.036**
Admission to intensive care unit	2 (5.7%)	18 (33.3%)	**0.002**	14 (60.9%)	4 (12.9%)	**<0.001**
Neonatal death 30 days	0	0	--	0	0	--
Fetal death	0	1 (1.9%)	1.000	0	1 (3.2%)	1.000
Other fetal events:	--	--	--	--	--	--
Time at Intensive Care Unit (among newborns requiring admission to ICU), days	4.5 (3–6)	10 (4–19)	0.379	4 (0–17)	0	**<0.001**
Time of hospitalization, days	3 (2–3)	5 (3–20)	**<0.001**	21 (13–28)	4 (3–4)	**<0.001**

Legend for Table 3: Data are expressed as absolute and relative frequencies (n,%) or median (25th–75th percentile). ICU, intensive care unit.

**Table 4 ijms-26-06684-t004:** Biomarkers in the first trimester of pregnancy in women who later had or did not have severe adverse events.

	Women Without Severe Adverse Events	Women with Severe Adverse Events	*p*-Value
	n = 82	n = 7	--
sFlt-1, pg/mL	1278.0 [1049.0–1752.0]	2126.0 [1922.0–2325.0]	**0.009**
PlGF, pg/mL	28.9 [20.4–40.9]	29.6 [19.1–38.2]	0.819
sFlt-1/PlGF	46.0 [35.0–56.2]	54.1 [40.9–99.0]	0.148
cfDNA, ng/mL	1010.0 [648.0–1725.0]	1965.0 [1680.0–3080.0]	**0.041**
Total MVs, (U/µL)	4862.3 [2517.1–9345.0]	5410.2 [2795.5–13,841.2]	0.745
Endothelial MVs, (U/µL)	34.1 [14.3–78.9]	80.4 [14.2–4597.2]	0.471
Platelet MVs, (U/µL)	812.3 [403.9–1575.2]	1073.5 [436.8–7697.6]	0.576
β-hCG, ng/mL	40.8 [29.3–65.8]	81.4 [37.3–94.7]	**0.048**
PAPP-A, mUI/mL	1.2 [0.6–2.3]	3.2 [1.0–6.0]	**0.046**
Total microvesicles AV^+^; endothelial microvesicles AV^+^ CD31^+^ CD41^−^; platelet microvesicles AV^+^ CD31^+^ CD41^+^.			

Legend for Table 4: Data are expressed as mean ± SD or median (25th–75th percentile). AV, annexin V; β-hCG, beta subunit of human chorionic gonadotropin; cfDNA, cell-free DNA; MVs, microvesicles; PAPP-A, pregnancy-associated plasma protein A; PlGF, placental growth factor; sFlt-1, soluble FMS-like tyrosine kinase-1.

## Data Availability

The datasets generated and/or analyzed during the current study are not publicly available due to ethical and legal restrictions. Specifically, the informed consent obtained from participants and the approval granted by the institutional ethics committee did not include provisions for public or third-party data sharing. Making the data available in a public repository would require re-consenting participants, which is not feasible.

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
