# Peer review of "Association of First-Trimester Maternal Biomarkers with Preeclampsia and Related Maternal and Fetal Severe Adverse Events"

_ijms, 2025, doi:10.3390/ijms26146684_

Round 1

Reviewer 1 Report

Comments and Suggestions for Authors

In this manuscript, authors assessed the association between known (PlGF, sFlt-1, betaHCG, PAPPA) and novel (cell-free DNA; total, endothelial and platelet microvesicles -MVs-) maternal blood biomarkers measured at first trimester with a later diagnosis of preeclampsia and with the development of severe adverse events (SAE).

The manuscript is interesting and generally well written. However, several points deserve to be improved. See my comments below.

Lines 63-66: An appropriate introduction of PE pathophysiology is necessary. In particular, PE risk factors, inflammation and metabolic dysfunctions (PMID: 38681101) characterizing this pathology deserve to be mentioned. 

Line 75: ??

Lines 89-93: other first trimester biomarkers have been evaluated (see PMID: 38892323) 

Tables: I would write the statistically significant p values in bold

Figures 2-4: Images are difficult to read since they are too small

Figures 2 and 4: I suggest to put the AUC values in the images

References: authors must follow journal style

Author Response

In this manuscript, authors assessed the association between known (PlGF, sFlt-1, betaHCG, PAPPA) and novel (cell-free DNA; total, endothelial and platelet microvesicles -MVs-) maternal blood biomarkers measured at first trimester with a later diagnosis of preeclampsia and with the development of severe adverse events (SAE).

The manuscript is interesting and generally well written. However, several points deserve to be improved. See my comments below.

Comments 1: Lines 63-66: An appropriate introduction of PE pathophysiology is necessary. In particular, PE risk factors, inflammation and metabolic dysfunctions (PMID: 38681101) characterizing this pathology deserve to be mentioned. 

Response 1: We thank the reviewer for this suggestion. We have expanded the introduction to briefly explain the pathophysiology of preeclampsia, including the main risk factors, the role of inflammation, and metabolic dysfunctions, as recommended.

We added the following text to the introduction:

PE is a multifactorial and complex disorder characterized by abnormal placentation, endothelial dysfunction, metabolic dysfunctions and systemic inflammatory response11.Several clinical risk factors for the development of PE have been identified, including a history of PE in prior gestations, chronic hypertension, diabetes, obesity, and advanced maternal age.

Comments 2: Line 75: ??

Response 2: Thank you for spotting this mistake. It seems that an automatic entrance of the word processor was added. This has now been corrected and the text “Final del formulario” has been removed here and elsewhere in the document.

Comments 3: Lines 89-93: other first trimester biomarkers have been evaluated (see PMID: 38892323) 

Response 3: We thank the reviewer for this suggestion. We have now cited the reference PMID: 38892323 in this paragraph as reference number 14 in the new version of the manuscript.

Comments 4: Tables: I would write the statistically significant p values in bold

Response 4: We appreciate this suggestion and have updated all tables to present statistically significant p-values in bold font for better clarity.

Comments 5: Figures 2-4: Images are difficult to read since they are too small

Response 5: Thank you for your feedback regarding Figures 2-4. We have carefully reviewed the image quality and resolution of these figures, and confirmed that they have been prepared at optimal resolution according to the journal instructions. The figures display clearly when viewed at appropriate magnification levels in the PDF format. We appreciate your attention to the visual clarity of our work and are committed to ensuring all figures meet the journal's publication standards.

Comments 6: Figures 2 and 4: I suggest to put the AUC values in the images

Response 6: Thank you for this suggestion. We have reviewed Figures 2 and 4 to incorporate the AUC values within the figure panels.

Comments 7: References: authors must follow journal style

Response 7: Thank you. We have reviewed and updated all references to ensure full compliance with the journal’s citation style.

Reviewer 2 Report

Comments and Suggestions for Authors

  1. The authors present their findings on a retrospective case-control study analyzing clinical and biochemical variables and exploring if there are associations between these variable and preeclampsia (early or late-onset) as well as maternal and fetal adverse events.  Because of the study design, only associations can be observed - not prediction.  In case-control studies such as this the OR or adjusted OR should be presented.  
  2. There is no power analysis and the numbers of cases and controls appear to be limited by the available samples and not by any sample size calculations.  This should be discussed in the limitations/weaknesses of the study
  3. 43 pregnancies were excluded due to no sample available – this introduces a major source of bias.  This needs to be accounted for the statistical analysis and discussed as a limitation
  4. Please explain how the patients were identified and enrolled to begin with? As part of a larger study? as part of a prospective data collection repository?
  5. how were the pregnancies enrolled and identified to begin with?
  6. Because the AUC curves generated include clinical variables such as primiparity, IVF, multifetal gestations, and higher BP, the data on AUC should be presented with the with the clinical variables alone and then with the addition of the biochemical tests.  In other words, how much do the biochemical tests add to the AUC? And is that difference significant or not.
  7. The use of ASA was not uniform.  Twelve patients were on ASA, others not.  This needs to be addressed in the Discussion.  how might that influence the results.  
  8. The association between maternal SAEs and sFlt-1 and cfDNA 0.92 may be confounded by their association with preeclampsia.  This needs to be accounted for statistically in the analysis.  
  9. The discussion of clinical implications is premature.  There are no clinical implications yet.  That section needs to be removed.  If these associations are validated in a prospective cohort using prespecified cutoffs then there may be some clinical implications.  
  10. Please confirm that patients underwent consent prospectively at time of blood draws, and that this is a secondary analysis of an existing cohort. 
  11. The retrospective nature of the trial is not a strength because the diagnosis are known. Even in prospective trials the outcomes must be clearly identified and analysis only occurs after deliveries are completed.  In fact using a retrospective data set is a huge source of bias because the authors are reliant on a previous investigator for making the diagnosis and entering the data correctly.  There is also the concern about how to handle missing cases and incomplete data cases.   This needs to be controlled in the statistical analysis.  
  12. There is a mention regarding the microvesicles that the processing times for microvesicles were not standard and that this may have impacted their accuracy.  This needs to be discussed in more detail since it is one of the few novel findings of this manuscript.
  13. The authors report that cfDNA, sFlt-1 and microvesicles were associated with late-onset PET and maternal SAEs.  however in Table 2 it is the sFlt-1/PIGF ratio not sFlt-1 that is significant, and in Table 4 the microvesicles are not significantly associated with severe maternal SAEs
  14. The microvesicle data is interesting and novel.  Consider limiting the analysis and discussion to the microvesicles or microvesicles and cfDNA.  The assocations with PIGF and PAPPA and preeclampsia are not novel . 
Comments on the Quality of English Language

  1. in the title of Table 4 – there is a word "latterly" - I think the authors mean "subsequently"? if that is correct please change
  2. In the Discussion the authors state that the patients were recruited after delivery.  I hope that is not the case, since blood was drawn in the first trimester.  Perhaps the authors mean that the patients were identified after delivery, when the outcomes are known.  This is the definition of a case-control study.  Please calrify.

Author Response

Comments 1. The authors present their findings on a retrospective case-control study analyzing clinical and biochemical variables and exploring if there are associations between these variable and preeclampsia (early or late-onset) as well as maternal and fetal adverse events.  Because of the study design, only associations can be observed - not prediction.  In case-control studies such as this the OR or adjusted OR should be presented.  

Response 1. We thank the reviewer for these comments and recommendations. Accordingly, the following manuscript implementations have been performed:

  1. a) Study design and terminology– we have revised the manuscript to replace terms such as “prediction” or “predictive” with “association” or “associated” wherever they refer to our own findings. We have, however, retained the original wording in a few instances where we discuss the findings of other studies or where a conditional phrasing (e.g., “may” or “could”) is used to suggest possible future research directions or clinical implications beyond the scope of our current dataset and findings.

  1. b) Use of odds ratios (ORs): We now provide odds ratios (ORs) with 95% confidence intervals for variables associated with adverse maternal or fetal outcomes, including the main OR results in the text and full numbers in the Supplementary Table S1, as follows:

“In addition to ROC curve analysis, we explored the strength of association between key biomarkers and adverse outcomes using odds ratios (ORs) with 95% confidence intervals. PLGF levels >36 pg/mL were associated with a significantly lower odds of fetal adverse events compared to levels ≤36 pg/mL (OR: 0.25, 95% CI: 0.09–0.69; p=0.007). A similar association was observed when considering combined maternal and fetal adverse events (OR: 0.27, 95% CI: 0.10–0.70; p = 0.008).  On the other hand, elevated sFLT1 concentrations (>1858 pg/mL) were associated with an increased OR of maternal adverse events (OR: 29.14, 95% CI: 3.25–261.38; p=0.003). There was also a non-significant trend toward higher maternal odds with cfDNA levels >1162 ng/mL (OR: 8.06, 95% CI: 0.93–69.99; p = 0.059). No statistically significant associations were found between cfDNA or sFLT1 levels and fetal or combined outcomes, nor between PLGF and maternal adverse events. The full results for the association of ORs with SAEs is provided in Supplementary Table S1.”

We would like to emphasize that the primary aim of our study was not to estimate the magnitude of risk, but to assess the discriminative performance of first-trimester biomarkers in distinguishing between pregnancies that did and did not later develop preeclampsia or related complications. This was evaluated through receiver operating characteristic (ROC) curve analysis and area under the curve (AUC) values, which are appropriate and widely accepted methods in retrospective case-control studies.In this context, measures such as AUC, sensitivity and specificity provide clinically meaningful information regarding the potential utility of these biomarkers in early screening frameworks. We believe that combining both approaches —reporting ORs and evaluating discriminative ability—provides a more complete and rigorous assessment of the biomarkers under study.

Comments 2. There is no power analysis and the numbers of cases and controls appear to be limited by the available samples and not by any sample size calculations.  This should be discussed in the limitations/weaknesses of the study

AND

Comments 3. 43 pregnancies were excluded due to no sample available – this introduces a major source of bias.  This needs to be accounted for the statistical analysis and discussed as a limitation

Response 2 and 3. We thank the reviewer for these remarks.

  1. a) Sample size calculation– A formal power analysis was in fact performed based on data from a previous study of our group. We have now included a detailed description of the sample size calculation in the Materials and Methods section (page 14, paragraph 2). In brief, assuming a clinically relevant difference of 10% in mean total circulating DNA between groups, with 80% power and a 5% significance level, a total of 142 subjects were required (82 with preeclampsia and 60 controls). This calculation also accounted for an estimated 50% sample loss, which unfortunately materialized due to sample handling and storage limitations.

  1. b) Missing samples and potential bias– As noted, 43 pregnancies were excluded due to unavailability of first-trimester biological samples. These losses were not selective by outcome and resulted primarily from technical or logistical issues in sample retrieval and processing. Nevertheless, we acknowledge this as a relevant limitation that may introduce selection bias and impact generalizability, as pertinently suggested by the reviewer. We have expanded the explanation of this limitation in the revised version (Discussion, page 12, last paragraph) to more explicitly address its potential implications.

Comments 4. Please explain how the patients were identified and enrolled to begin with? As part of a larger study? as part of a prospective data collection repository? how were the pregnancies enrolled and identified to begin with?

Response 4. We thank the reviewer for the opportunity to clarify the enrollment process. All participants were identified and recruited after delivery, during their postpartum hospital stay. Women who met the inclusion criteria were informed about the study and asked for written informed consent for the use of their clinical data and access to stored biological samples. These samples had been previously collected and biobanked during a routine first-trimester antenatal visit, which is standard practice for all pregnant women in our healthcare setting. This approach allowed us to retrospectively analyze first-trimester biomarkers in pregnancies with known outcomes, while ensuring ethical standards and proper patient consent. We have now added a more detailed description of the recruitment and consent process to the Materials and Methods section (page 13, paragraph 3).

Comments 5. Because the AUC curves generated include clinical variables such as primiparity, IVF, multifetal gestations, and higher BP, the data on AUC should be presented with the with the clinical variables alone and then with the addition of the biochemical tests. In other words, how much do the biochemical tests add to the AUC? And is that difference significant or not.

Response 5. We appreciate the reviewer’s thoughtful suggestion, which addresses an important point in the evaluation of the additive value of biomarkers. We have now explored the performance of models with clinical variables alone versus models that also included biomarkers. However, we would like to note the following:

  • One of the multivariable models (maternal SAE) included no clinical variables —only cfDNA and sFlt-1—so no comparison could be made in this case.
  • Two models (overall PE and fetal SAE) included only one clinical variable(primiparity and multiple gestation, respectively), all of which are binary variables. Since ROC curves cannot be meaningfully constructed for single dichotomous predictors, we could not generate ROC curves for the clinical component alone in those cases.

For the late-onset PE model, we were able to construct a ROC curve using the clinical variable alone (systolic blood pressure –continuous variable), which yielded an AUC of 0.61 (95% CI: 0.49–0.74, p=0.059). This modest and non-significant discriminative ability supports the added value of platelet-derived microvesicles, which, when included in the model, substantially increased the AUC to 0.81 (95% CI: 0.71–0.92).

Two models —early-onset PE and maternal-fetal SAE— included two or more clinical variables and were suitable for comparative ROC analysis.

  • In the early-onset PE model, a ROC curve using only the clinical variables (primiparity and IVF) yielded an AUC of 77 (95% CI: 0.65–0.89). This was notably lower than the AUC of 0.89 (95% CI: 0.82–0.96)obtained from the full model including PlGF and PAPP-A, indicating that the inclusion of these biomarkers substantially improved the model’s discriminative performance in this scenario.
  • In the maternal-fetal SAEmodel, after removing PlGF, maternal age was no longer a significant predictor, leaving only IVF (a binary variable). As previously noted, this consideration prevented us the generation of a meaningful ROC curve for the clinical model alone.

We have revised the corresponding parts of the Results and Discussion sections to clarify which models demonstrate added value from biomarkers, and which do not. We emphasize the contribution of biomarkers where they clearly improve model performance (e.g., late-onset PE), and appropriately downplay their contribution where their incremental value is minimal (e.g., early-onset PE). In addition, we have included a short paragraph in the Statistical Analysis section of the Materials and Methods (page 15, paragraph 5) to account for these new analyses.

Comments 6. The use of ASA was not uniform.  Twelve patients were on ASA, others not.  This needs to be addressed in the Discussion.  how might that influence the results. 

Response 6. We agree with the reviewer that the non-uniform use of acetylsalicylic acid (ASA) should be acknowledged. In our cohort, ASA was prescribed based on clinical criteria in accordance with standard obstetric guidelines, primarily in patients with high-risk profiles. We have now addressed this point in the Discussion (page 12 paragraph 3). Importantly, in our analysis, ASA use was not significantly associated with the development of preeclampsia or with adverse maternal or fetal outcomes. Thus, while its use reflects clinical heterogeneity, it did not appear to have a major impact in our findings.

Comments 7. The association between maternal SAEs and sFlt-1 and cfDNA 0.92 may be confounded by their association with preeclampsia.  This needs to be accounted for statistically in the analysis. 

Response 7. We appreciate this comment. However, we respectfully believe that adjusting for preeclampsia (PE) in the analysis of first-trimester biomarkers and severe maternal adverse events (SAEs) would not be appropriate for two main reasons:

  1. Temporal sequence– All biomarkers and clinical variables tested for association were measured in the first trimester, whereas the diagnosis of PE —and consequently, the occurrence of maternal complications— took place later in pregnancy. As such, PE is not a confounder in this context, but rather a potential mediator in the causal pathway between early biomarkers and adverse outcomes.
  2. Study objective– A central aim of our study was to explore which early biomarkers may help stratify the risk of severe outcomes among pregnancies that go on to develop PE. Adjusting for PE would obscure this clinical objective and potentially introduce overadjustment bias.

We have now clarified this reasoning in the revised Discussion section (page 11, paragraph 4), where we explicitly address the role of PE as a downstream event rather than a confounding variable in our analysis.

Comments 8. The discussion of clinical implications is premature.  There are no clinical implications yet.  That section needs to be removed.  If these associations are validated in a prospective cohort using prespecified cutoffs then there may be some clinical implications.  

Response 8. We thank the reviewer for this important point. We agree that, given the retrospective design and exploratory nature of our findings, it is premature to discuss clinical implications at this stage. Accordingly, we have removed the “Clinical implications” subtitle and adapt the content in the Discussion in the revised manuscript. We have included a more cautious statement emphasizing the need for prospective validation of our results in independent cohorts before any clinical application can be considered.

Comments 9. Please confirm that patients underwent consent prospectively at time of blood draws, and that this is a secondary analysis of an existing cohort. 

Response 9. We appreciate the opportunity to clarify this point. First-trimester blood samples used in this study were originally collected during routine antenatal care and stored in the institutional biobank. This is standard practice for all pregnant women in our healthcare setting (not driven by our study). Patients were invited to participate in our study after delivery, during their postpartum hospitalization, at which point written informed consent was obtained for the use of their clinical data and access to their first-trimester stored samples. Therefore, this study constitutes a retrospective analysis based on biobanked specimens and clinical data from an existing cohort, with full ethical approval and patient consent obtained prior to any research use.

Comments 10. The retrospective nature of the trial is not a strength because the diagnosis are known. Even in prospective trials the outcomes must be clearly identified and analysis only occurs after deliveries are completed.  In fact using a retrospective data set is a huge source of bias because the authors are reliant on a previous investigator for making the diagnosis and entering the data correctly.  There is also the concern about how to handle missing cases and incomplete data cases. This needs to be controlled in the statistical analysis.  

Response 10. We thank the reviewer for this thoughtful comment. We agree that the retrospective design inherently limits causal inference and introduces potential sources of bias. Accordingly, we have removed any reference to the retrospective nature of the study as a strength in the revised Discussion section.

Regarding the quality of the data, we would like to clarify that all clinical and obstetric information was obtained from electronic health records routinely used in clinical practice. These data are collected prospectively by attending clinicians and include variables essential for medical decision-making, thus offering a high degree of reliability and completeness. Importantly, these are the same clinical data upon which real-world diagnostic and therapeutic decisions are based, we do not use different or complex clinical information that had to be collected for the purpose of the study. In contrast, biomarker data were obtained ad hoc for this study, using first-trimester samples stored in the institutional biobank. This distinction is now clarified in the revised Methods section.

Concerning missing data, only cases with complete data for the relevant variables were included in each analysis. We performed complete-case analyses and have now specified this explicitly in the Methods section with this new subsection: “4.4. Data Sources and Quality Control” (page 14, paragraph 3). We also acknowledge in the Discussion that missing samples and the retrospective design represent limitations of our study.

Comments 11. There is a mention regarding the microvesicles that the processing times for microvesicles were not standard and that this may have impacted their accuracy. This needs to be discussed in more detail since it is one of the few novel findings of this manuscript.

Response 11. We thank the reviewer for pointing out this comment and agree that further clarification is warranted. As noted, the blood samples used for microvesicles (MV) analysis were collected and processed under routine clinical conditions, without strict control over processing times. While this may have affected the absolute quantification of MVs, all samples (cases and controls) were handled under the same biobanking protocols, ensuring internal consistency and comparability between groups.

Our aim was not to define reference ranges or absolute concentrations of MVs, but rather to evaluate their potential discriminatory performance in a real-world clinical setting. The fact that MV levels showed differential associations despite these limitations suggests potential robustness, which merits further investigation under standardized conditions.

We have elaborated on this point in the revised Discussion section (page 13, paragraph 1), and now acknowledge both the limitation and its implications for the interpretation of MV-related findings.

Comments 12. The authors report that cfDNA, sFlt-1 and microvesicles were associated with late-onset PET and maternal SAEs.  however in Table 2 it is the sFlt-1/PIGF ratio not sFlt-1 that is significant, and in Table 4 the microvesicles are not significantly associated with severe maternal SAEs

Response 12. We appreciate the reviewer’s observation and agree that our previous phrasing was too general. We have revised the text in the Discussion section to more precisely reflect the specific associations observed in our study. In particular:

  • For late-onset PEplatelet-derived microvesicleswere the best biomarker showing a statistically significant association, while cfDNA showed a non-significant association with low predictive value.
  • For maternal severe adverse events, both sFlt-1and cfDNA were associated with the outcome, with AUCs of 0.80 and 0.73, respectively.

The sFlt-1/PlGF ratio, while showing significance in univariate analysis, was not included in the final multivariate models for maternal SAEs. The corresponding sentence in the Conclusions section has been updated for clarity and accuracy:

Interestingly, our findings reveal that placental biomarkers, such as PlGF and PAPP-A, are more closely associated with early-onset PE and fetal complications, whereas markers of endothelial dysfunction and cell stress are more closely linked to maternal adverse outcomes (cfDNA and sFlt-1) and late-onset PE (platelet microvesicles).

Comments 13. The microvesicle data is interesting and novel.  Consider limiting the analysis and discussion to the microvesicles or microvesicles and cfDNA.  The assocations with PIGF and PAPPA and preeclampsia are not novel. 

Response 13. We sincerely thank the reviewer for highlighting the novelty and relevance of the microvesicles and cfDNA data. While we agree that this represents one of the most innovative aspects of our study, we have chosen to retain the analysis of previously studied biomarkers such as PlGF and PAPP-A, particularly in areas where their clinical value remains uncertain or inconsistently reported in the literature —such as their role in late-onset PE or in predicting adverse maternal or fetal outcomes.

Including these well-established markers also allowed us to build multivariable models that combine both novel and validated biomarkers, better reflecting the potential real-world integration of emerging tools into current risk stratification frameworks. Moreover, it provides an internal benchmark to interpret the performance of the newer biomarkers.

Comments 14. In the title of Table 4 – there is a word "latterly" - I think the authors mean "subsequently"? if that is correct please change

Response 14. Thanks for the suggestion. We have made the change.

Comments 15. In the Discussion the authors state that the patients were recruited after delivery.  I hope that is not the case, since blood was drawn in the first trimester.  Perhaps the authors mean that the patients were identified after delivery, when the outcomes are known.  This is the definition of a case-control study. 

Response 15. Thank you for this clarification. We apologize for the confusion in our initial description. It is included in previous answers to suggestions, but to clarify our study design: Our healthcare system routinely performs first trimester visits for all pregnant women, primarily for chromosomal screening. During these visits, blood samples are collected and stored for 12-18 months for potential reanalysis when clinically indicated. For this study, we identified and recruited participants after delivery. We then obtained informed consent from participants to use their previously collected first trimester blood samples and clinical data for research purposes. We have reviewed the Methods section to clarify this point. Please also see responses to suggestions 4 and 7.

Round 2

Reviewer 1 Report

Comments and Suggestions for Authors

the manuscript has been significantly improved and can be accepted in the present form 

Author Response

Comment: the manuscript has been significantly improved and can be accepted in the present form 

Response: We sincerely thank Reviewer 1 for their positive evaluation and kind feedback. We are glad that the revised manuscript meets their expectations and appreciate their support throughout the review process.